# Central Kisspeptin Does Not Affect ERK1/2 or p38 Phosphorylation in Oxytocin Neurons of Late-Pregnant Rats

**DOI:** 10.3390/ijms23147729

**Published:** 2022-07-13

**Authors:** Mehwish Abbasi, Rachael A. Augustine, Karl J. Iremonger, Colin H. Brown

**Affiliations:** 1Brain Health Research Centre, University of Otago, Dunedin 9054, New Zealand; abbme833@student.otago.ac.nz (M.A.); rachael.augustine@otago.ac.nz (R.A.A.); karl.iremonger@otago.ac.nz (K.J.I.); 2Centre for Neuroendocrinology, University of Otago, Dunedin 9054, New Zealand; 3Department of Physiology, School of Biomedical Sciences, University of Otago, Dunedin 9054, New Zealand

**Keywords:** kisspeptin, oxytocin, supraoptic nucleus, paraventricular nucleus, pregnancy

## Abstract

Oxytocin is secreted by hypothalamic supraoptic nucleus (SON) and paraventricular nucleus (PVN) oxytocin neurons to induce uterine contractions during parturition. Increased activation of oxytocin neurons at parturition involves a network of afferent inputs that increase oxytocin neuron excitability. Kisspeptin fibre density increases around oxytocin neurons during pregnancy, and central kisspeptin administration excites oxytocin neurons only in late pregnancy. Kisspeptin signals via extracellular regulated kinase 1/2 (ERK1/2) and p38. Therefore, to determine whether kisspeptin excites oxytocin neurons via ERK1/2-p38 signalling in late-pregnant rats, we performed immunohistochemistry for phosphorylated ERK1/2 (pERK1/2) and phosphorylated p38 (p-p38) in oxytocin neurons of non-pregnant and late-pregnant rats. Intracerebroventricular (ICV) kisspeptin administration (2 µg) did not affect pERK1/2 or p-p38 expression in SON and PVN oxytocin neurons of non-pregnant or late-pregnant rats. Furthermore, ICV kisspeptin did not affect pERK1/2 or p-p38 expression in brain areas with major projections to the SON and PVN: the nucleus tractus solitarius, rostral ventrolateral medulla, locus coeruleus, dorsal raphe nucleus, organum vasculosum of the lamina terminalis, median preoptic nucleus, subfornical organ, anteroventral periventricular nucleus, periventricular nucleus and arcuate nucleus. Hence, kisspeptin-induced excitation of oxytocin neurons in late pregnancy does not appear to involve ERK1/2 or p38 activation in oxytocin neurons or their afferent inputs.

## 1. Introduction

The hormone oxytocin induces uterine contractions for delivery of the offspring during birth. Although oxytocin is not essential for birth [1], it is necessary for the normal progression of birth because oxytocin receptor antagonism delays the onset of, and prolongs the duration of, parturition in rats [2]. Oxytocin is synthesised by hypothalamic supraoptic nucleus (SON) and paraventricular nucleus (PVN) oxytocin neurons that each send a single axon to the posterior pituitary gland, where oxytocin secretion is triggered by action potential invasion of the axon terminal [3]. Increased activation of oxytocin neurons at parturition is triggered by afferent inputs that relay peripheral signals from cervical stretch receptors [4]. The best characterised excitatory afferent input to oxytocin neurons arises from the A2 noradrenergic cell group of the nucleus tractus solitarius (NTS), which is robustly activated at parturition [5]. However, acute activation of central noradrenergic receptors alone is not sufficient to trigger parturition in late-pregnant rats [6], indicating that other mechanisms are likely also involved.

Kisspeptin neurons also project to the SON [7], and kisspeptin fibre density increases around oxytocin neurons during pregnancy [8]. Furthermore, intracerebroventricular (ICV) kisspeptin excites oxytocin neurons only in late pregnancy [8], and we have recently shown that this excitation is mediated by direct effects on oxytocin neurons as well as by enhancement of excitatory afferent signalling [9].

Kisspeptin principally signals via kisspeptin receptor 1 (Kiss1R) [10] but also has high affinity for neuropeptide FF receptors (NPFFR) [11]. Classically, Kiss1R activation increases phosphorylation of extracellular regulated kinase 1/2 (ERK1/2) [10,12,13,14,15] but also increases phosphorylation of p38 in some cell lines [13,14]. Similarly to Kiss1R, NPFFR activation also increases phosphorylation of ERK1/2 [16,17,18,19] and p38 [19].

Although ERK1/2 and p38 signalling pathways share upstream regulators [20], p38 phosphorylation has been observed without measurable changes in ERK1/2 phosphorylation, suggesting that p38 can also be activated via a different, as yet unidentified, pathway [21,22]. Furthermore, phosphorylated p38 (p-p38) can directly suppress ERK1/2 phosphorylation [23].

Although kisspeptin excites oxytocin neurons, at least in part, by a direct effect [9], it is unknown whether this excitation is mediated by activation of the canonical Kiss1R signalling pathway. Therefore, to test the hypothesis that kisspeptin directly activates oxytocin neurons at the end of pregnancy via ERK1/2 and/or p38 signalling, we measured phosphorylated ERK1/2 (pERK1/2) and p-p38 expression in oxytocin neurons after ICV kisspeptin administration to non-pregnant and late-pregnant rats. Although intra-SON kisspeptin directly excites oxytocin neurons at the end of pregnancy, this effect is not robust and likely does not underpin the full kisspeptin-mediated excitation [9]. Therefore, to determine whether afferent inputs might be activated by kisspeptin to excite oxytocin neurons at the end pregnancy, we also measured pERK1/2 and p-p38 expression in brain areas that express Kiss1R and/or NPFFR and project to the SON and/or PVN. We found that ICV kisspeptin did not affect pERK1/2 or p-p38 expression in oxytocin neurons or any of the brain areas examined, suggesting that kisspeptin activates oxytocin neurons at the end of pregnancy via an alternative signalling pathway.

## 2. Results

### 2.1. Kisspeptin Does Not Affect pERK1/2 Expression in SON or PVN Oxytocin Neurons

We labelled for pERK1/2 and oxytocin in the SON and PVN of non-pregnant and gestation day 21 (G21, the expected day of parturition) rats, and example photomicrographs are shown in Figure 1a–d,h–k. Although there was a higher proportion of oxytocin-positive neurons co-expressing pERK1/2 in the SON of aCSF-treated (32.56 ± 4.18%, *n* = 7) and kisspeptin-treated (25.62 ± 3.19%, *n* = 8) G21 rats than in aCSF-treated (11.44 ± 4.18%, *n* = 8) and kisspeptin-treated (7.40 ± 1.13%, *n* = 8) non-pregnant rats, there was no effect of ICV kisspeptin on the proportion of oxytocin-positive neurons co-expressing pERK1/2 in the SON of non-pregnant and G21 rats (Reproductive status (RS): F_(1,1)_ = 33.17, *p* ˂ 0.0001; Treatment (T): F_(1,1)_ = 4.20, *p* = 0.050; RS × T: F_(1,2)_ = 0.050, *p* = 0.824, two-way ANOVA; Figure 1e–g).

Similarly, there was a higher proportion of oxytocin-positive neurons co-expressing pERK1/2 in the PVN of aCSF-treated (42.71 ± 5.70%, *n* = 7) and kisspeptin-treated (41.25 ± 4.45%, *n* = 8) G21 rats than in aCSF-treated (20.33 ± 4.69%, *n* = 9) and kisspeptin-treated (19.11 ± 3.29%, *n* = 9) non-pregnant rats, but there was no effect of ICV kisspeptin on the proportion of oxytocin-positive neurons co-expressing pERK1/2 in the PVN of non-pregnant and G21 rats (RS: F_(1,1)_ = 28.52, *p* ˂ 0.0001; T: F_(1,1)_ = 0.0009, *p* = 0.975; RS × T: F_(1,2)_ = 0.039, *p* = 0.844, two-way ANOVA; Figure 1l–n).

Subdividing the PVN into the magnocellular PVN (mPVN) and parvocellular PVN (pPVN) revealed similar results for both regions as was found for the PVN as whole. There was higher proportion of oxytocin-positive neurons co-expressing pERK1/2 in the mPVN of aCSF-treated (46.93 ± 6.71%, *n* = 7) and kisspeptin-treated (43.14 ± 4.95%, *n* = 8) G21 rats than in the aCSF-treated (22.04 ± 5.37%, *n* = 9) and kisspeptin-treated (19.10 ± 3.12%, *n* = 9) non-pregnant rats, but there was no effect of ICV kisspeptin on the proportion of oxytocin-positive neurons co-expressing pERK1/2 in the mPVN (RS: F_(1,1)_ = 26.27, *p* < 0.0001; T: F_(1,1)_ = 0.028, *p* = 0.866; RS × T: F_(1,2)_ = 0.0006, *p* = 0.980; Figure 2a–c) of non-pregnant and G21 rats. Similarly, there was higher proportion of oxytocin-positive neurons co-expressing pERK1/2 in the pPVN of aCSF-treated (37.09 ± 4.99%, *n* = 7) and kisspeptin-treated (37.37 ± 4.75%, *n* = 8) G21 rats than in aCSF-treated (17.68 ± 4.04%, *n* = 9) and kisspeptin-treated (18.24 ± 3.81%, *n* = 9) non-pregnant rats, but there was no effect of ICV kisspeptin on the proportion of oxytocin-positive neurons co-expressing pERK1/2 in the pPVN (RS: F_(1,1)_ = 23.15, *p* < 0.0001; T: F_(1,1)_ = 0.076, *p* = 0.783; RS × T: F_(1,2)_ = 0.186, *p* = 0.669; Figure 2d–f) of non-pregnant and G21 rats.

### 2.2. Kisspeptin Does Not Affect p-p38 Expression in SON or PVN Oxytocin Neurons

We labelled for p-p38 and oxytocin in the SON and PVN of non-pregnant and G21 rats, and example photomicrographs are shown in Figure 3a–d,h–k. Similarly to pERK1/2, there was a higher proportion of oxytocin-positive neurons co-expressing p-p38 in the SON of aCSF-treated (34.96 ± 2.64%, *n* = 5) and kisspeptin-treated (37.62 ± 3.01%, *n* = 7) G21 rats than in aCSF-treated (28.79 ± 3.04%, *n* = 8) and kisspeptin-treated (26.21 ± 3.58%, *n* = 9) non-pregnant rats, but there was no effect of ICV kisspeptin on the proportion of oxytocin-positive neurons co-expressing p-p38 in the SON of non-pregnant or G21 rats (RS: F_(1,1)_ = 9.90, *p* = 0.004; T: F_(1,1)_ = 0.007, *p* = 0.930; RS × T: F_(1,2)_ = 0.052, *p* = 0.821, two-way ANOVA; Figure 3e–g). By contrast to the SON, there was no effect of reproductive status or ICV kisspeptin on p-p38 expression in oxytocin neurons in the PVN as a whole (RS: F_(1,1)_ = 3.39, *p* = 0.077; T: F_(1,1)_ = 0.939, *p* = 0.341; RS × T: F_(1,2)_ = 4.23, *p* = 0.050; Figure 3l–n) or in the mPVN (RS: F_(1,1)_ = 2.90, *p* = 0.100; T: F_(1,1)_ = 0.99, *p* = 0.32; RS × T: F_(1,2)_ = 3.55, *p* = 0.071; Figure 4a–c) or pPVN (RS: F_(1,1)_ = 3.35, *p* = 0.078; T: F_(1,1)_ = 0.353, *p* = 0.557; RS × T: F_(1,2)_ = 3.35, *p* = 0.078; Figure 4d–f) of non-pregnant (aCSF-treated, *n* = 8, kisspeptin-treated, *n* = 8) and G21 rats (aCSF-treated, *n* = 5, kisspeptin-treated, *n* = 8).

### 2.3. Kisspeptin Does Not Affect pERK1/2 or p-p38 Expression in Brainstem Noradrenergic Neurons in the NTS, RVLM or LC

To determine whether ICV kisspeptin might excite oxytocin neurons at the end of pregnancy by activation of NTS noradrenergic neurons, which project to SON and PVN oxytocin neurons, express NPFFR [24] and are robustly activated at parturition [5], pERK1/2 and p-p38 were each double-labelled with tyrosine hydroxylase (TH, a marker for identification of noradrenaline-containing neurons). pERK1/2 and p-p38 were also each double-labelled with TH in the rostral ventrolateral medulla (RVLM) to determine whether excitatory effects of ICV kisspeptin might be mediated via RVLM because RVLM noradrenergic neurons are also activated at parturition [5], express Kiss1R [25] and project catecholaminergic axons (containing dopamine, noradrenaline, adrenaline) to SON and PVN oxytocin (and vasopressin) neurons [26].

Examples of pERK1/2 and TH labelling in the NTS and RVLM of non-pregnant and G21 rats are shown in Figure 5a–d,h–k. There was no effect of reproductive status or ICV kisspeptin on the number of TH-positive neurons co-expressing pERK1/2 in the NTS (RS: F_(1,1)_ = 0.580, *p* = 0.453; T: F_(1,1)_ = 0.501, *p* = 0.485; RS × T (F_(1,2)_ = 0.900, *p* = 0.351, two-way ANOVA; Figure 5e–g) of non-pregnant (aCSF-treated, *n* = 7, kisspeptin-treated, *n* = 8) and G21 rats (aCSF-treated, *n* = 7, kisspeptin-treated, *n* = 8). Irrespective of kisspeptin treatment, pERK1/2 expression in TH-positive NTS neurons correlated with pERK1/2 expression in oxytocin-positive SON neurons (r = 0.536, *p* = 0.039), but there was no correlation between pERK1/2 expression in TH-positive NTS neurons and oxytocin-positive PVN neurons (r = 0.098, *p* = 0.726) or mPVN neurons (r = 0.221, *p* = 0.428) or pPVN neurons (r = −0.192, *p* = 0.492) in G21 rats.

Although there was a higher number of TH-positive neurons co-expressing pERK1/2 in the RVLM of aCSF-treated G21 rats (*n* = 6) than aCSF-treated non-pregnant rats (*n* = 8), there was no effect of ICV kisspeptin on the number of TH-positive neurons co-expressing pERK1/2 in the RVLM of non-pregnant or G21 rats (RS: F_(1,1)_ = 5.27, *p* = 0.029; T: F_(1,1)_ = 0.667, *p* = 0.421; RS × T: F_(1,2)_ = 6.35, *p* = 0.018; Figure 5l–n). Additionally, there was no correlation between pERK1/2 expression in TH-positive RVLM neurons and oxytocin-positive SON neurons (r = 0.063, *p* = 0.836) in G21 rats.

Examples of p-p38 and TH labelling in the NTS and RVLM of non-pregnant and G21 rats are shown in Figure 6a–d,h–k. Similarly to pERK1/2, there was no effect of reproductive status or ICV kisspeptin on the number of TH-positive neurons co-expressing p-p38 in the NTS (RS: F_(1,1)_ = 0.086, *p* = 0.771; T: F_(1,1)_ = 0.344, *p* = 0.562; RS × T: F_(1,2)_ = 0.086, *p* = 0.771; Figure 6e–g) of non-pregnant (aCSF-treated, *n* = 7, kisspeptin-treated, *n* = 7) and G21 rats (aCSF-treated, *n* = 7, kisspeptin-treated, *n* = 8). Additionally, there was no correlation between p-p38 expression in TH-positive NTS neurons and oxytocin-positive SON neurons (r = −0.287, *p* = 0.392), PVN neurons (r = 0.181, *p* = 0.572), mPVN neurons (r = 0.169, *p* = 0.598) or pPVN neurons (r = 0.198, *p* = 0.536) in G21 rats. By contrast to pERK1/2 expression in TH-positive RVLM neurons, there was no effect of reproductive status or ICV kisspeptin on p-p38 expression in TH-positive RVLM neurons (RS: F_(1,1)_ = 1.44, *p* = 0.241; T: F_(1,1)_ = 0.005, *p* = 0.944; RS × T: F_(1,2)_ = 0.005, *p* = 0.944; Figure 6l–n) of non-pregnant (aCSF-treated, *n* = 6, kisspeptin-treated, *n* = 8) and G21 rats (aCSF-treated, *n* = 6, kisspeptin-treated, *n* = 6). Additionally, there was no correlation between p-p38 expression in TH-positive RVLM neurons and oxytocin-positive SON neurons (r = −0.199, *p* = 0.607) in G21 rats.

pERK1/2 was labelled in the locus coeruleus (LC) to serve as a brain area control for the NTS because the LC expresses Kiss1R and NPFFR [24,27], and LC noradrenergic neurons project to the PVN but principally project to non-magnocellular neurons [28,29]. Examples of pERK1/2 labelling in the LC of non-pregnant and G21 rats are shown in Figure 1a–d. As expected, there was no effect of reproductive status or ICV kisspeptin on the number of pERK1/2-positive neurons in the LC (RS: F_(1,1)_ = 1.19, *p* = 0.286; T: F_(1,1)_ = 0.563, *p* = 0.460; RS × T: F_(1,2)_ = 0.292, *p* = 0.594; Figure 1e) of non-pregnant (aCSF-treated, *n* = 7, kisspeptin-treated, *n* = 7) and G21 rats (aCSF-treated, *n* = 5, kisspeptin-treated, *n* = 8). Additionally, there was no correlation between pERK1/2 expression in the LC and oxytocin-positive neurons of SON in G21 rats (r = −0.266, *p* = 0.401).

### 2.4. Kisspeptin Does Not Affect pERK1/2 Expression in the DRN Neurons

Dorsal raphe nucleus (DRN) neurons express Kiss1R and NPFFR [24,30] and project to SON and PVN oxytocin neurons [31] but DRN inputs to oxytocin neurons are not involved in parturition [32]. Therefore, pERK1/2 was labelled in the DRN to serve as brainstem control for inputs to oxytocin neurons that are activated at parturition. Examples of pERK1/2 labelling in the DRN of non-pregnant and G21 rats are shown in Figure 1f–i. Although there was a higher number of pERK1/2-positive neurons in the DRN of G21 rats than non-pregnant rats, there was no effect of ICV kisspeptin on the number of pERK1/2-positive neurons in the DRN of non-pregnant (aCSF-treated, *n* = 8, kisspeptin-treated, *n* = 6) or G21 rats (aCSF-treated, *n* = 4, kisspeptin-treated, *n* = 7) (RS: F_(1,1)_ = 4.38, *p* = 0.048; T: F_(1,1)_ = 0.077, *p* = 0.783; RS × T: F_(1,2)_ = 0.015, *p* = 0.901, two-way ANOVA; Figure 1j). Additionally, there was no correlation between pERK1/2 expression in DRN neurons and SON oxytocin-positive neurons in G21 rats (r = −0.522, *p* = 0.099).

### 2.5. Kisspeptin Does Not Affect pERK1/2 or p-p38 Expression in the OVLT, MnPO or SFO

To determine whether ICV kisspeptin activates organum vasculosum of the lamina terminalis (OVLT), median preoptic nucleus (MnPO) and/or subfornical organ (SFO) neurons, which each project to the SON and PVN [33,34,35] and express Kiss1R and/or NPFFR [25,27,30,36], pERK1/2 and p-p38 were labelled in each brain area.

Examples of pERK1/2 labelling in the OVLT, MnPO and SFO of non-pregnant and G21 rats are shown in Figure 7a–d,f–i,k–n. There was no effect of reproductive status or ICV kisspeptin on the number of pERK1/2-positive OVLT neurons in non-pregnant (aCSF-treated, *n* = 8, kisspeptin-treated, *n* = 6) and G21 rats (aCSF-treated, *n* = 4, kisspeptin-treated, *n* = 5) (RS: 0.194, *p* = 0.664; T: F_(1,1)_ = 0.021, *p* = 0.884; RS × T: F_(1,2)_ = 1.62, *p* = 0.218, two-way ANOVA; Figure 7e), MnPO neurons in non-pregnant (aCSF-treated, *n* = 7, kisspeptin-treated, *n* = 7) and G21 rats (aCSF-treated, *n* = 6, kisspeptin-treated, *n* = 6) (RS: F_(1,1)_ = 1.63, *p* = 0.215; T: F_(1,1)_ = 0.003, *p* = 0.956; RS × T: F_(1,2)_ = 0.0009, *p* = 0.976; Figure 7j) or SFO neurons in non-pregnant (aCSF-treated, *n* = 6, kisspeptin-treated, *n* = 6) and G21 rats (aCSF-treated, *n* = 5, kisspeptin-treated, *n* = 5) (RS: F_(1,1)_ = 0.00001, *p* = 0.999; T: F_(1,1)_ = 0.035, *p* = 0.851; RS × T: F_(1,2)_ = 0.073, *p* = 0.789; Figure 7o). Additionally, there was no correlation between pERK1/2 expression in oxytocin-positive SON neurons and pERK1/2 expression in OVLT neurons (r = −0.065, *p* = 0.866), MnPO neurons (r = −0.259, *p* = 0.415) or SFO neurons (r = −0.105, *p* = 0.772) in G21 rats.

Examples of p-p38 labelling in the OVLT, MnPO and SFO of non-pregnant and G21 rats are shown in Figure 8a–d,f–i,k–n. Similarly to pERK1/2, there was no effect of reproductive status or ICV kisspeptin on the number of p-p38-positive OVLT neurons in non-pregnant (aCSF-treated, *n* = 8, kisspeptin-treated, *n* = 5) and G21 rats (aCSF-treated, *n* = 4, kisspeptin-treated, *n* = 4) (RS: F_(1,1)_ = 0.601, *p* = 0.448; T: F_(1,1)_ = 0.231, *p* = 0.636; RS × T: F_(1,2)_ = 0.019, *p* = 0.890; Figure 8e), MnPO neurons in non-pregnant (aCSF-treated, *n* = 4, kisspeptin-treated, *n* = 5) and G21 rats (aCSF-treated, *n* = 4, kisspeptin-treated, *n* = 3) (RS: F_(1,1)_ = 0.00007, *p* = 0.993; T: F_(1,1)_ = 0.003, *p* = 0.952; RS × T: F_(1,2)_ = 0.408, *p* = 0.534; Figure 8j) or SFO neurons in non-pregnant (aCSF-treated, *n* = 6, kisspeptin-treated, *n* = 4) and G21 rats (aCSF-treated, *n* = 3, kisspeptin-treated, *n* = 5) (RS: F_(1,1)_ = 1.31, *p* = 0.271; T: F_(1,1)_ = 1.50, *p* = 0.240; RS × T: F_(1,2)_ = 1.32, *p* = 0.268, Figure 8o). Additionally, there was no correlation between p-p38 expression in oxytocin-positive SON neurons and pERK1/2-positive OVLT neurons (r = 0.112, *p* = 0.809), MnPO neurons (r = −0.648, *p* = 0.115) or SFO neurons (r = 0.189, *p* = 0.684) in G21 rats.

### 2.6. Kisspeptin Does Not Affect pERK1/2 or p-p38 Expression in the AVPe, PeN or pERK1/2 Expression in the ARC

pERK1/2 and p-p38 were also labelled in the anteroventral periventricular nucleus (AVPe), periventricular nucleus (PeN) and arcuate nucleus (ARC), which contain kisspeptin neurons [37,38], express Kiss1R and/or NPFFR [25,30,36]. Kisspeptin PeN neurons project to the SON but AVPe and ARC kisspeptin neurons do not [8].

Examples of pERK1/2 labelling in the AVPe, PeN and ARC of non-pregnant and G21 rats are shown in Figure 9a–d,f–i,k–n. There was no effect of reproductive status or ICV kisspeptin on the number of pERK1/2-positive AVPe neurons in non-pregnant (aCSF-treated, *n* = 8, kisspeptin-treated, *n* = 5) and G21 rats (aCSF-treated, *n* = 4, kisspeptin-treated, *n* = 3) (RS: F_(1,1)_ = 0.270, *p* = 0.609; T: F_(1,1)_ = 0.126, *p* = 0.726; RS × T: F_(1,2)_ = 0.090, *p* = 0.767, two-way ANOVA; Figure 9e), PeN neurons in non-pregnant (aCSF-treated, *n* = 3, kisspeptin-treated, *n* = 3) and G21 rats (aCSF-treated, *n* = 4, kisspeptin-treated, *n* = 4) (RS: F_(1,1)_ = 0.046, *p* = 0.834; T: F_(1,1)_ = 0.159, *p* = 0.698; RS × T: F_(1,2)_ = 0.498, *p* = 0.496; Figure 9j) or ARC neurons in non-pregnant (aCSF-treated, *n* = 9, kisspeptin-treated, *n* = 9) and G21 rats (aCSF-treated, *n* = 6, kisspeptin-treated, *n* = 8) (RS: F_(1,1)_ = 0.764, *p* = 0.389; T: F_(1,1)_ = 0.719, *p* = 0.403; RS × T: F_(1,2)_ = 0.358, *p* = 0.554; Figure 9o). Also, there was no correlation between pERK1/2 expression in oxytocin-positive SON neurons and pERK1/2-positive AVPe neurons (r = 0.069, *p* = 0.882), PeN neurons (r = −0.143, *p* = 0.735) or ARC neurons (r = −0.074, *p* = 0.809) in G21 rats.

Examples of p-p38 labelling in the AVPe and PeN of non-pregnant and G21 rats are shown in Figure 10a–d,f–i. Similarly to pERK1/2, there was no effect of reproductive status or ICV kisspeptin on the number of p-p38-positive AVPe neurons in non-pregnant (aCSF-treated, *n* = 7, kisspeptin-treated, *n* = 5) and G21 rats (aCSF-treated, *n* = 3, kisspeptin-treated, *n* = 4) (RS: F_(1,1)_ = 0.231, *p* = 0.637; T: F_(1,1)_ = 0.882, *p* = 0.362; RS × T: F_(1,2)_ = 0.225, *p* = 0.641; Figure 10e) or PeN neurons in non-pregnant (aCSF-treated, *n* = 4, kisspeptin-treated, *n* = 6) and G21 rats (aCSF-treated, *n* = 4, kisspeptin-treated, *n* = 4) (RS: F_(1,1)_ = 0.455, *p* = 0.510; T: F_(1,1)_ = 0.0008, *p* = 0.977; RS × T: F_(1,2)_ = 0.559, *p* = 0.467; Figure 10j). Additionally, there was no correlation between p-p38 expression in oxytocin-positive SON neurons and pERK1/2-positive AVPe neurons (r = −0.378, *p* = 0.459) or PeN neurons (r = −0.178, *p* = 0.672) in G21 rats. There was insufficient tissue to label p-p38 in the ARC.

## 3. Discussion

We have recently shown that kisspeptin excites oxytocin neurons in late pregnancy, in part, by a direct action on oxytocin neurons [9]. The present study revealed that ICV kisspeptin did not induce ERK1/2 or p38 phosphorylation in oxytocin neurons of non-pregnant or late-pregnant rats when administered at a dose that increases oxytocin neuron firing rate in late-pregnant rats [8]. Nevertheless, consistent with previous findings [39], ERK1/2 (and p38) phosphorylation was higher in oxytocin neurons of late-pregnant rats than in non-pregnant rats. Hence, it appears likely that kisspeptin excitation of oxytocin neurons in late-pregnant rats is not mediated by Kiss1R and/or NPFFR activation of ERK1/2 or p38 signalling.

### 3.1. Local Kisspeptin Activation of Oxytocin Neurons in Late Pregnancy

The failure of kisspeptin to induce ERK1/2 (or p38) phosphorylation in oxytocin neurons of late-pregnant rats probably does not reflect a failure to deliver sufficient kisspeptin to activate these signalling pathways because kisspeptin administration increased ERK1/2 phosphorylation in gonadotrophin-releasing hormone (GnRH) neurons of the same non-pregnant rats in which ERK1/2 and p38 phosphorylation were unchanged in oxytocin neurons. Although kisspeptin did not induce ERK1/2 phosphorylation in GnRH neurons of late-pregnant rats, this likely reflects downregulation of GnRH neuron responsiveness to kisspeptin during pregnancy, as we have previously reported using Fos protein as a marker of activation [40]. Elevated ERK1/2 phosphorylation in oxytocin neurons at the end of pregnancy might have occluded the ability of exogenous kisspeptin to induce further phosphorylation. However, ≤50% of oxytocin neurons expressed pERK1/2 in late-pregnant rats. Hence, occlusion also appears unlikely to account for the failure of kisspeptin to induce ERK1/2 phosphorylation in oxytocin neurons of late-pregnant rats. Taken together, these observations suggest that the lack of kisspeptin-induced ERK1/2 (and p38) phosphorylation in oxytocin neurons in late pregnancy was not due to a technical failure. Rather, it appears that kisspeptin excitation of oxytocin neurons in late pregnancy is not mediated by Kiss1R-mediated phosphorylation of ERK1/2 or p38.

Direct kisspeptin excitation of oxytocin neurons results from transiently increased excitability after each action potential (post-spike excitability) [9]. Post-spike excitability is underpinned by an afterdepolarisation triggered by each action potential [41]. The afterdepolarisation is more prominent in oxytocin neurons in late-pregnant rats than in non-pregnant rats [42]. Kisspeptin excites GnRH neurons by activating non-specific cation channels and inhibiting potassium channels [43], which underpin the afterdepolarisation [44,45]. Hence, the direct kisspeptin-induced enhancement of post-spike excitability of oxytocin neurons in late-pregnant rats might be mediated by increasing the afterdepolarisation. However, the mechanism by which kisspeptin enhances the post-spike excitability of oxytocin neurons remains to be determined.

If kisspeptin does not activate Kiss1R/NPFFR-ERK1/2/p38 signalling in oxytocin neurons, another signalling pathway must mediate the direct effects of kisspeptin on oxytocin neurons in late pregnancy. Alternatively, or additionally, Kiss1R activates neuronal nitric oxide synthase in preoptic neurons via phosphatidylinositol-3-kinase(PI3K)-Akt [46]. Hence, PI3K-Akt might mediate the direct effects of kisspeptin on oxytocin neurons in late pregnancy. Finally, kisspeptin activation of oxytocin neurons might not be mediated by Kiss1R. Indeed, SON Kiss1R mRNA expression does not change during pregnancy [8], suggesting that upregulation of Kiss1R does not underpin kisspeptin excitation of oxytocin neurons in late pregnancy, although the possibility of increased Kiss1R surface expression and/or sensitivity cannot be discounted. Oxytocin neurons also express NPFFR [47], which couples to Gs and Gi as well as to Gq [24,48,49]. NPFFR activation stimulates cAMP production in olfactory bulb neurons [50] and increases phosphorylation of cAMP response element-binding protein (CREB) and c-Jun N terminal kinase (JNK) as well as activating c-Jun [19]. Hence, NPFFR-CREB/JNK signalling might mediate the direct effects of kisspeptin on oxytocin neurons in late pregnancy. pCREB/JNK expression could be investigated in further work to determine which, if any, of these signalling pathways mediate kisspeptin excitation of oxytocin neurons at the end of pregnancy.

### 3.2. Lack of Kisspeptin Induction of ERK1/2 or p38 Phosphorylation in Afferent Inputs to the Oxytocin System

Although our recent findings showed that kisspeptin excites oxytocin neurons in late pregnancy, in part, by a direct action of kisspeptin on oxytocin neurons, our findings also suggested that activation of afferent inputs might also contribute to the excitation [9]. Therefore, we mapped kisspeptin-induced ERK1/2 and p38 phosphorylation in brain areas that project to the oxytocin system and express Kiss1R and NPFFR (LC, DRN, OVLT, SFO, AVPe, ARC) [25,27,30,36], only NPFFR (NTS, PeN) [36] or only Kiss1R (RVLM, MnPO) [25,51]. However, neither ERK1/2 nor p38 phosphorylation was affected by kisspeptin in any of the brain areas studied in non-pregnant or late-pregnant rats. Hence, it appears that any indirect kisspeptin excitation of oxytocin neurons in late pregnancy is not mediated by activation of ERK1/2-p38 signalling in these afferent inputs.

Although kisspeptin did not induce ERK1/2 or p38 phosphorylation in afferent inputs to the oxytocin system in non-pregnant or late-pregnant rats, there was a positive correlation of ERK1/2 phosphorylation in late pregnancy between SON oxytocin neurons and NTS noradrenergic neurons, which project to oxytocin neurons and are activated during parturition [5]; the correlation with ERK1/2 phosphorylation in oxytocin neurons was specific to NTS noradrenergic neurons, further implicating NTS noradrenergic neurons in driving oxytocin neuron activity for parturition.

Although the same caveats apply to the lack of ERK1/2 or p38 phosphorylation in their afferent inputs as apply to the oxytocin neuron themselves, a further possibility is that kisspeptin might enhance excitatory synaptic transmission via presynaptic actions on oxytocin neurons afferent inputs. If active, this mechanism does not involve local glutamatergic or GABAergic inputs to oxytocin neurons, which are unaffected by kisspeptin in brain slices from non-pregnant and late-pregnant rats [9]. Therefore, the most likely candidate might be the NTS noradrenergic input, which is activated at parturition [5], releasing noradrenaline [32] to excite oxytocin neurons via α_1_-adrenoreceptors [4]. Additionally, there is a possibility that kisspeptin excites oxytocin neurons via dopaminergic inputs. Dopamine synapses are evident on oxytocin neurons in the SON [52], and dopamine levels rise within the SON during the second phase of parturition [32]. Furthermore, dopamine directly activates neurons via dopamine D2 receptors [53] and facilitates synchronized burst firing of oxytocin neurons during lactation [54]. Hence, dopaminergic inputs are also a potential candidate by which kisspeptin could excite oxytocin neurons in late-pregnant rats.

## 4. Materials and Methods

### 4.1. Ethical Approval

All experimental procedures were approved by the University of Otago Animals Ethics Committee (approval number: AUP-19-44) and carried out in accordance with the New Zealand Animal Welfare Act and associated guidelines.

### 4.2. Animals

Adult female Sprague-Dawley rats (6–12 weeks of age) were purchased from the University of Otago Animal Facility and housed in controlled temperature and lighting (22–24 °C; 12 h light/12 h dark), with free access to standard laboratory rodent food and water. Non-pregnant rats were freely-cycling virgin rats and were housed in groups of 3–5 until after surgery. Primiparous pregnant rats were used on G21..

For timed mating, the oestrous cycle stage was assessed by vaginal cytology. At pro-oestrus, rats were placed overnight with a male for mating and the next morning were considered to be G0 after confirmation of the presence of sperm in the vaginal smear.

### 4.3. Intracerebroventricular Cannulation and Kisspeptin Administration

Under isoflurane anaesthesia (2/2.5% in 1 L min^−1^ O_2_), non-pregnant and G13/14 rats were implanted with an ICV guide cannula (22-gauge, Plastics One Inc., Roanoke, VA, USA) into the lateral cerebral ventricle (co-ordinates relative to bregma, in mm: right lateral = 1.3; rostral/caudal = 0: ventral = 3.0) using stereotaxic surgical procedures, as previously described [55]. The guide cannula was anchored to screws (1 mm) inserted into the dorsal surface of the skull using light cure adhesive transbond (Henry Schein Shalfoon, Auckland, New Zealand). All rats were individually housed post-surgery. The clinical condition of each rat was monitored daily until the day of experiment.

On day seven or eight following surgery, non-pregnant and G21 rats were anaesthetised with intraperitoneal pentobarbitone (60 mg kg^−1^; 300 mg mL^−1^). Upon cessation of the flexor withdrawal reflex, an internal cannula (28-guage, Plastics One Inc., Roanoke, VA, USA) was inserted into the lateral cerebral ventricle through the guide cannula and attached to a Hamilton syringe (2 µL, Hamilton Company, Reno, CF, USA) via a polyethylene tube (PE-10, 0.5 mm diameter). Rats were injected ICV with either aCSF (artificial cerebrospinal fluid) or 2 µg kisspeptin (1 µg µL^−1^ dissolved in aCSF, Merck, Kenilworth, NJ, USA) over 1 min, 60 min after induction of anaesthesia.

### 4.4. Brain Collection and Sectioning

At 15 min after ICV kisspeptin or aCSF administration, rats were perfused transcardially with ~50 mL of 0.9% saline followed by 250–300 mL of 4% paraformaldehyde in 0.1 M phosphate buffer (pH 7.6) under continued pentobarbitone anaesthesia. This timeframe was selected because kisspeptin-induced pERK1/2 and p-p38 levels peak 10–15 min after incubation with kisspeptin in various cell lines [13,14,56]. Brains were post-fixed for 24 h in 4% paraformaldehyde solution and then 30% sucrose in 0.1 M phosphate buffer (pH 7.6) at 4 °C for 72 h.

Then, 30 µm coronal sections were cut on a freezing microtome (Leica SM2400, Wetzlar, Germany) from +0.60 to −3.00 mm relative to bregma to capture the SON, PVN, medial septum (MS), OVLT, MnPO, SFO, AVPe, PeN and ARC in the forebrain, from −7.56 to −8.04 mm for the DRN, −9.60 to −10.08 mm for the LC and from −12.00 to −14.64 mm for the NTS and RVLM in the brainstem [57]. Sections containing forebrain areas were collected in 4 series, and brainstem areas were collected in 3 series; each series contained 6–8 sections. Sections were stored in fresh cryoprotectant (pH 7.6, 0.05 M phosphate buffer saline, 0.9% sodium chloride, 30% sucrose, 1% polyvinylpyrrolidone, 30% ethylene glycol) at −20 °C until use.

### 4.5. Immunohistochemistry

For fluorescent immunohistochemistry, sections were washed in Tris-buffered saline (TBS) and endogenous aldehydes were blocked by incubating in 0.1% sodium borohydride in TBS for 20 min at room temperature (RT). Sections were washed in TBS and placed in incubation solution (0.3% Triton X−100 and 0.25% bovine serum albumin in TBS) containing host serum in which secondary antibody was raised for 60 min to avoid non-specific binding. Next, sections were incubated in incubation solution containing a cocktail of primary antibodies for 48 h on an orbital shaker. After washing in TBS, sections were incubated for 3 h at RT in fluorescent-tagged secondary antibodies diluted in incubation solution. Primary antibodies used for fluorescent staining of pERK1/2 and p-p38 with oxytocin and TH were rabbit anti-pERK1/2 antibody (p-44/42 MAPK (T 202/Y 204), 9101S, Cell Signalling; 1:1000), rabbit anti-p-p38 antibody (Thr 180/Tyr, 182, 9211, Cell Signalling; 1:2000), mouse oxytocin (MAB-5296, Millipore, Burlington, MA, USA; 1:5000) and mouse anti-TH (MAB 318, Millipore, Burlington, MA, USA; 1:2000). The secondary antibodies used were goat anti-mouse Alexa fluor 488 antibody (A11029, Thermofisher Scientific, Waltham, MA, USA; 1:500) and goat anti-rabbit Alexa fluor 568 antibody (ab 175471, Abcam, Cambridge, UK; 1:500).

For chromogenic immunohistochemistry, sections were washed in Tris-buffered saline (TBS) and endogenous aldehydes were blocked by incubating in TBS containing methanol and 30% H_2_O_2_ for 10–15 min at RT. Brain sections were washed and incubated in rabbit anti-pERK1/2 (1: 5000) or rabbit anti-p-p38 (1:2000) antibody with 4% normal goat serum for 48 h. Sections were then incubated in a secondary antibody solution containing a goat anti-rabbit biotinylated secondary antibody (BA-1000, Vector Laboratories, Birmingham, AL, USA; 1:500) for 90 min. After washing again in TBS, sections were incubated in an avidin-biotin-peroxidase solution (Vector Laboratories, Newark, CA, USA) for 90 min. Finally, pERK1/2-p-p38 staining was visualised by immersing sections in a nickel-diaminobenzidine (Ni DAB) peroxidase solution (Vector Laboratories, Newark, CA, USA) prepared in distilled water. Some sections were randomly mounted on slides and regularly examined under the bright field microscope. When staining became visible against background (usually after 2–10 min), the Ni DAB reaction was stopped by washing the sections in TBS.

Sections were mounted on gelatinized slides, cleared, dehydrated (only DAB-stained sections pass through increasing concentrations of ethanol series followed by xylene) and dried. Fluorescent-stained sections were coverslipped using Fluoromout-G^®^ (Southern Biotech, Birmingham, AL, USA), whereas DAB-stained sections were coverslipped using DPX mounting medium (VWR International Limited, Poole, UK). Stained sections were examined using an Olympus bright-field/fluorescence microscope (BX51- NAOS) attached to a digital camera (GRYPHAX) to capture photomicrographs. The brain areas of interest were photographed at magnifications of 500×, 200× and 100×. Quantification was completed manually using the *cell counter* plugin on Fiji (Image J, National Institute of Health, v. 1.47, Bethesda, MD, USA) software. From each animal, 3–6 sections per brain area were used to calculate the mean number of positive neurons per group. For all analyses, slides were randomly coded and quantified by an experimenter blinded to the code to avoid bias in counting.

Activation of pERK1/2 by 2 µg ICV kisspeptin was confirmed by double labelling for GnRH and pERK1/2 in the MS. First, DAB staining of pERK1/2 was performed as described above, and sections were co-stained using guinea pig anti-GnRH primary antibody (GA02, a generous gift from Professor Allan Herbison; 1:10,000) with 4% normal goat serum for 48 h. Sections were then incubated in secondary antibody solution containing a goat anti-guinea pig (BA-1000, Vector Laboratories, Newark, CA, USA; 1:500). Finally, the DAB solution without nickel was added to label GnRH-expressing neurons. Examples of pERK1/2 and GnRH labelling in the MS of non-pregnant and G21 rats are shown in Figure 2a–d. As expected, the proportion of GnRH neurons expressing pERK1/2 was significantly higher in kisspeptin-treated non-pregnant rats (90.15 ± 3.89%, *n* = 4) than in aCSF-treated non-pregnant rats (36.32 ± 11.5%, *n* = 6) but was not different between kisspeptin-treated late-pregnant rats (53.67 ± 11.1%, *n* = 8) and aCSF-treated late-pregnant rats (52.77 ± 9.05% *n* = 6) (RS: F_(1,1)_ = 1.064, *p* = 0.314; T: F_(1,1)_ = 4.461, *p* = 0.047; RS × T interaction: F_(1,2)_ = 6.77, *p* = 0.017, two-way ANOVA; Figure 2g), which was a similar outcome to our previous study using Fos protein as a marker of activation [40].

Previous studies validated the specificity of oxytocin [58], pERK1/2 and p-p38 [59], TH [60] and GnRH [61] antibodies. Specificity controls were performed by omitting primary antibodies. No non-specific staining was evident in any section with primary antibody omitted.

### 4.6. Statistical Analysis

Data were analysed on GraphPad Prism version 8 for Windows (GraphPad Software Inc., San Diego, CA, USA). Statistical significance between groups was determined by two-way analysis of variance (ANOVA) followed by *post hoc* Holm-Sidak’s test, where the F ratio was significant. All values are presented as mean ± standard error of mean (SEM). Pearson product moment correlations were run to determine correlations. Probabilities (*p*) < 0.05 were considered significant.

### 4.7. Concluding Remarks

Taken together, the current results show that kisspeptin-induced excitation of oxytocin neurons in late pregnancy is not mediated by phosphorylation of canonical Kiss1R (or NPFFR)-activated second messengers, ERK1/2 (or p38), and further work will be required to determine which signalling pathway mediates kisspeptin excitation of oxytocin neurons in late pregnancy.

## Data Availability

The data presented in this study are available on request from the corresponding author. The data are not publicly available.

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
