# Peer review of "Central Kisspeptin Does Not Affect ERK1/2 or p38 Phosphorylation in Oxytocin Neurons of Late-Pregnant Rats"

_ijms, 2022, doi:10.3390/ijms23147729_

Round 1

Reviewer 1 Report

The research of the authors and others is well designed and the results are theoretically coherent, and we rate this paper as a good paper worthy of publication. Some minor questions are listed below.

1) "NPFFR-CREB/JNK signaling may mediate the direct action of kisspeptin on oxytocin neurons in late pregnancy. I find the author's point about "which of these signaling pathways mediates the excitation of kisspeptin on oxytocin neurons in late pregnancy" very interesting. I would like to see more depth of speculation regarding the point at which to consider which specific signaling pathways might be involved.

2) Although kisspeptin neurons may affect several neurons during pregnancy and lactation via dopamine neurons in the dorsal arcuate nucleus, is it possible that dopamine neurons may be involved in this study?

3) Are there any changes in the actual number of oxytocin neurons or kisspeptin receptors that are not related to ERK1/2 or p38 phosphorylation?

Author Response

1) "NPFFR-CREB/JNK signaling may mediate the direct action of kisspeptin on oxytocin neurons in late pregnancy. I find the author's point about "which of these signaling pathways mediates the excitation of kisspeptin on oxytocin neurons in late pregnancy" very interesting. I would like to see more depth of speculation regarding the point at which to consider which specific signaling pathways might be involved.

We have no data to underpin further speculation on which pathways might be involved in direct kisspeptin effects but have made minor revisions in the text of discussion regarding this point (lines 455-461):

“NPFFR activation stimulates cAMP production in olfactory bulb neurons [50], and in-creases phosphorylation of cAMP response element-binding protein (CREB) and c-Jun N terminal kinase (JNK) as well as activating c-Jun [19]. Hence, NPFFR-CREB/JNK signalling might mediate the direct effects of kisspeptin on oxytocin neurons in late pregnancy. pCREB/JNK expression could be investigated in further work to determine which, if any, of these signalling pathways mediate kisspeptin excitation of oxytocin neurons at the end of pregnancy.”

2) Although kisspeptin neurons may affect several neurons during pregnancy and lactation via dopamine neurons in the dorsal arcuate nucleus, is it possible that dopamine neurons may be involved in this study?

Yes, there is the possibility that kisspeptin excites oxytocin neurons via dopaminergic inputs and this is addressed in the text of discussion section, with the insertion of the following text (lines 489 – 496):

“Additionally, there is a possibility that kisspeptin excites oxytocin neurons via dopa-minergic inputs. Dopamine synapses are evident on oxytocin neurons in the SON [52] and dopamine levels rise within the SON during the second phase of parturition [32]. Furthermore, dopamine directly activate neurons via dopamine D2 receptors [53] and facilitates synchronized burst firing of oxytocin neurons during lactation [54]. Hence, dopaminergic inputs are also a potential candidate by which kisspeptin could excite oxytocin neurons in late-pregnant rats.”

3) Are there any changes in the actual number of oxytocin neurons or kisspeptin receptors that are not related to ERK1/2 or p38 phosphorylation?

We did not observe any changes in the number of oxytocin neurons and results are stated in the legends of Figure 1 and 2.

Reviewer 2 Report

The manuscript is well-written, insightful, and of interest to the readers.

Author Response

The manuscript is well-written, insightful, and of interest to the readers.

No response required.

Reviewer 3 Report

SUMMARY

                Kisspeptin is a neuropeptide that signals through Kisspeptin 1 receptor and neuropeptide FF receptors and is an established regulator of the mammalian reproductive axis. Kisspeptin neurons project to oxytocinergic neurons and excite them in late pregnancy. However, the molecular mechanisms by which kisspeptin regulates oxytocinergic neurons is currently unknown. These data show that although kisspeptin has been shown to regulate p-ERK and P38, that is not the mechanism associated with excitation of oxytocinergic neurons.  Overall, the manuscript is well written and the data appears convincing, pending confirmation of an appropriate number of animals used for each group and an appropriate replication of the experiments. Moreover, negative data can be as important as positive data and is all too often not published. However, without confirmation of the experimental numbers and replicates, I am unable to recommend publication until this information is provided.

Major points

11.       The authors do not state how many rats were used for each experiment per experimental group, nor how many times each experiment was replicated. For example in figure 1, I count approximately 8 circles for each bar e-g and l-n, which would suggest that these experiments were repeated. This information needs to be explicitly stated for each figure and supplementary figure.  

Minor points

22.       The authors start with Figure 1e-g in the Results text. Please explain 1a-d and 1h-k in the text of the paper in addition to the figure legend. This continues throughout the paper for figure 5a-d and 5h-k, 6a-d and 6h-k, supplementary figure 1a-d, and all other image figures.

Author Response

Major points

  1. The authors do not state how many rats were used for each experiment per experimental group, nor how many times each experiment was replicated. For example in figure 1, I count approximately 8 circles for each bar e-g and l-n, which would suggest that these experiments were repeated. This information needs to be explicitly stated for each figure and supplementary figure.

We apologise for this oversight on our part: the text of the results and figure legends are revised to include the number of rats used in experiments and these experiments were not repeated.

Minor points

  1. The authors start with Figure 1e-g in the Results text. Please explain 1a-d and 1h-k in the text of the paper in addition to the figure legend. This continues throughout the paper for figure 5a-d and 5h-k, 6a-d and 6h-k, supplementary figure 1a-d, and all other image figures.

The text of the results is revised to refer to Figure panels a-d and h-k.